# Effect of Doxycycline in Decreasing the Severity of *Clostridioides difficile* Infection in Mice

**DOI:** 10.3390/antibiotics11010116

**Published:** 2022-01-17

**Authors:** Bo-Yang Tsai, Yi-Hsin Lai, Chun-Wei Chiu, Chih-Yu Hsu, Yi-Hsuan Chen, Yueh-Lin Chen, Pei-Jane Tsai, Yuan-Pin Hung, Wen-Chien Ko

**Affiliations:** 1Institute of Basic Medical Sciences, College of Medicine, National Cheng Kung University, Tainan 704, Taiwan; anemoday@hotmail.com (B.-Y.T.); good4458@gmail.com (Y.-H.L.); 2Departments of Internal Medicine, Tainan Hospital, Ministry of Health & Welfare, Tainan 700, Taiwan; bahamudo@gmail.com; 3Department of Medical Laboratory Science and Biotechnology, College of Medicine, National Cheng Kung University, Tainan 704, Taiwan; hsudengsiang@gmail.com (C.-Y.H.); dkao3vm4@gmail.com (Y.-H.C.); box10533@gmail.com (Y.-L.C.); 4Center of Infectious Disease and Signaling Research, National Cheng Kung University, Tainan 704, Taiwan; 5Department of Pathology, National Cheng Kung University Hospital, College of Medicine, National Cheng Kung University, Tainan 704, Taiwan; 6Department of Internal Medicine, National Cheng Kung University Hospital, College of Medicine, National Cheng Kung University, Tainan 704, Taiwan; 7Department of Medicine, College of Medicine, National Cheng Kung University, Tainan 704, Taiwan

**Keywords:** *Clostridioides difficile*, doxycycline, exposure, mouse model, anti-inflammation, metronidazole, vancomycin

## Abstract

**Background:** Doxycycline possesses antibacterial activity against *Clostridioides difficile* and anti-inflammatory effects. **Materials and Methods:** The influence of doxycycline on the development of CDI was studied in an established animal model of CDI using C57BL/6 mice. **Results:** Mice intraperitoneally administered doxycycline had higher cecum weight (1.3 ± 0.1 vs. 0.5 ± 0.1 g; *p* < 0.001) and less body weight reduction (0.7 ± 0.5 g vs. −17.4 ± 0.2 g; *p* < 0.001) than untreated mice infected with *C. difficile*. Oral doxycycline, metronidazole, or vancomycin therapy resulted in less body weight reduction in mice with CDI than in untreated mice (1.1 ± 0.1 g, 1.3 ± 0.2 g, 1.2 ± 0.1 g, vs. 2.9 ± 0.3 g; *p* < 0.001). Doxycycline therapy led to lower expression levels of inflammatory cytokines, such as macrophage inflammatory protein-2 (0.4 ± 0.1 vs. 2.9 ± 1.3, *p* = 0.02), and higher levels of zonula occludens-1 (1.2 ± 0.1 vs. 0.8 ± 0.1, *p* = 0.02) in colonic tissues than in untreated mice. **Conclusions:** Concurrent intraperitoneal administration of doxycycline and oral *C. difficile* challenge does not aggravate the disease severity of CDI, and oral doxycycline may be a potential therapeutic option for CDI.

## 1. Introduction

*Clostridioides difficile* is one of the most common gastrointestinal pathogens, especially in nosocomial infections, with clinical symptoms ranging from mild diarrhea and pseudomembranous colitis to toxic megacolon, with a mortality of up to 25–40% [1,2,3,4,5,6,7,8,9]. Oral vancomycin and fidaxomicin have been suggested to replace metronidazole as therapeutic choices for either mild-moderate or severe *Clostridioides difficile* infection (CDI) in the clinical guidelines recommended by the Infectious Diseases Society of America (IDSA) and Society for Health Care Epidemiology of America (SHEA) in 2017 [10]. Vancomycin, besides a direct anti-*C. difficile* effect, had no anti-inflammatory effect [11]. In the mouse model of CDI, vancomycin therapy results in the mitigation of weight loss and diarrhea during acute infection but is associated with CDI recurrence and late-onset death, indicative of the need for other therapeutic agents [11]. However, administration of vancomycin plus an anti-inflammatory agent, for example, A2A adenosine receptor (A2AAR) agonist, decreased inflammation and improved survival rates [11]. In contrast, fidaxomicin and its primary metabolite OP-1118 were found to significantly inhibit *C. difficile* toxin A-mediated intestinal inflammation via inhibition of NF-κB Activity, and thus decreased tissue damage in the human colon [12]. Therefore, antibiotics with both and anti-*C. difficile* effect and anti-inflammation effect would be an excellent surrogate for treatment of CDI.

Doxycycline, a second-generation tetracycline antibiotic, has been shown to exhibit anti-inflammatory, antioxidant, antiapoptotic, neuroprotective, and immunomodulatory effects beyond antimicrobial effects [13]. Although the occurrence of CDI was noted after doxycycline prophylaxis for malaria [14], doxycycline was associated with a decreased incidence of CDI in recent studies [15,16,17]. A pharmacotherapy review concluded that doxycycline might have protective effects on the development of CDI based on evidence from retrospective studies [17]. The direct linkage of doxycycline exposure to CDI remains unresolved.

For clinical *C. difficile* isolates, the minimum inhibitory concentration (MIC) of doxycycline ranged from 0.016 μg/mL to ≥32 μg/mL with a resistance rate of 0–40% if resistance was defined as an MIC ≥ 8 μg/mL [18,19,20]. Higher doxycycline MICs were noted among some *C. difficile* isolates, especially the RT078 lineage isolates [20]. For *C. difficile* isolates with low doxycycline MICs, the clinical utility of doxycycline as a therapeutic option for CDI warrants further investigation. In this study, the potential role of doxycycline as a preventive or therapeutic choice for CDI was explored.

## 2. Materials and Methods

### 2.1. C. difficile Strain

A toxigenic *C. difficile* strain, VPI 10463 (CCUG 19126/ATCC 43255), belonging to ribotype 087 and toxinotype 0, without binary toxin, with a doxycycline MIC of 0.25 μg/mL was used in the animal experiments.

### 2.2. Concurrent Doxycycline Exposure and C. difficile Challenge in Mice

Six- to 8-week-old male C57BL/6 mice were exposed to an antibiotic cocktail, which included 0.045 mg/mL vancomycin, 0.215 mg/mL metronidazole, 0.4 mg/mL kanamycin, 0.035 mg/mL gentamicin, and 850 U/mL colistin, from 5 days to 1 day before oral inoculation of vegetative *C. difficile* bacteria [21] (Figure 1).

A proton pump inhibitor, esomeprazole (40 mg/kg/day), was added by oral gavage for 2 days [21]. In our published article about the mouse model of CDI [21], severe CDI was associated with frequent diarrhea, body weight loss, and less fecal content in cecum. The latter was reflected by a decrease in cecum weight. Therefore, in our study, body weight and cecum weight were measured to reflect the disease severity of CDI in the mouse model.

Each mouse received intraperitoneal administration of doxycycline (5 mg/kg) or control (vehicle: PBS, phosphate-buffered saline) and was concurrently challenged with vegetative cells of *C. difficile* in an oral inoculum of 3.5 × 10^7^ CFU at 3 h after antibiotic exposure (Figure 1) [22,23]. Uninfected mice were administrated with 5 days of an antibiotic cocktail but treated with PBS as the negative control group. All mouse studies with five mice in each group were repeated at least three times to confirm the results.

### 2.3. One Dose of Oral Antibiotic Therapy for CDI in Mice

Drinking water containing the antibiotic cocktail was administered from 5 days to 1 day and then esomeprazole (40 mg/kg/day) was given to mice orally, twice daily for 2 days before oral inoculation of *C. difficile* [21]. The mice were intraperitoneally administered clindamycin (4 mg/kg) and then challenged orally with *C. difficile* VPI 10463. After oral challenge with *C. difficile* for 24 h, one dose of doxycycline (125 mg/kg), metronidazole (250 mg/kg), vancomycin (50 mg/kg), or control (vehicle: PBS, phosphate-buffered saline) was administered by oral gavage (Figure 1) (i.p., intraperitoneally; PBS, phosphate-buffered saline; PPI, proton pump inhibitor).

All mouse studies with five mice in each group were repeated at least three times to confirm the results. This study was approved by IACUC of the National Cheng Kong University (Approval No. 102296).

### 2.4. Phenotypic Analysis of CDI

Reported signs of colitis in mice included weight loss and diarrhea [21,24]. Diarrhea was scored by stool consistency, as 0 = well-formed pellets, 1 = semiformed stools that did not adhere to the anus, 2 = semiformed stools that adhered to the anus, and 3 = liquid stools. Thus, the changes of body weight, stool consistency, gross view of gut, and cecal weight were selected to estimate the disease severity of CDI in mice [21,24].

### 2.5. Bacterial Burden

Bacterial loads of *C. difficile* in feces were measured to correlate with disease progression. To quantify bacterial loads, feces were collected and cultured. CFUs were counted in cycloserine-cefoxitin-fructose agar (CCFA) (CREATIVE LIFE SCIENCE CO., LTD., Taipei, Taiwan) at 37 °C in an incubator anaerobically for 2 days.

### 2.6. Expression of Inflammatory Cytokines and Tight Junction Proteins in Colonic Tissues

The mRNA expression levels of inflammatory cytokines, including interleukin [IL]-1β, macrophage inflammatory protein-2 (MIP-2, also known as C-X-C motif chemokine 2), IL-18, and tight junction proteins, including tight junction protein (TJP)-1, occludin (OCLN), and zonula occludens (ZO)-1, in colonic tissues were measured using quantitative reverse transcription polymerase chain reaction (qRT–PCR). Total RNA was extracted with REzol isolation reagent (Protech Technology Enterprise, Taipei, Taiwan), followed by affinity chromatography (RNeasy Plus Mini Kit) (QIAGEN, Hilden, Germany), and the complementary DNA (cDNA) of these samples was synthesized using M-MLV reverse transcriptase (Invitrogen, Waltham, MA, USA). To quantify the expression of genes, cDNA was mixed with SYBR green (Applied Biosystems, Waltham, MA, USA) and specific primers for the genes coding for IL-1β, CXCL-2, IL-18, Zo-1, and OCLN. PCR was run according to the standard protocol, and the emission of fluorescence was detected with StepOnePlus™ (Applied Biosystems, Waltham, MA, USA). Relative expression of the targeted genes was primarily normalized to a housekeeping gene, beta-actin (*ACTB*), followed by the 2^−^^Δct^ method analysis.

### 2.7. Histopathologic Examination

Mice were sacrificed 2 days after infection, and their colonic tissues were harvested for histopathologic examination. The colons were cut longitudinally and fixed with 10% formaldehyde (Mallinckrodt Pharmaceuticals, St Louis, MO, USA). The colonic tissue was embedded in paraffin and sectioned (5 µm) prior to hematoxylin and eosin staining (H&E stain). The stained, sectioned tissue was imaged through microscopy, and the integrity of the tissue was assessed. For pathological scoring, six fields per sample were examined and scored. Average counts of neutrophils in the six high-power fields (HPFs) in tissues were determined. The severity of colitis was scored, ranging between zero and three points for each of the following parameters: (1) polymorphonuclear infiltrate; (2) mononuclear infiltrate; (3) edema; (4) erosion and ulcerations; (5) crypt abscess; (6) crypt destruction; and (7) distribution of inflammation (mucosa = 1, mucosa and submucosa = 2, transmural inflammation = 3). Based on the total score (ranging from 0 to 21), the extent of tissue inflammation was graded as mild (1–5 points), moderate (6–10 points), or severe (>10 points) [25]. The markers of inflammation, including neutrophil infiltration, epithelial damage, and irregular shape of colon mucosa, were analyzed.

### 2.8. Statistical Method

Data were analyzed using the SPSS version 17.0.0 (IBM, Armonk, NY, USA), and Prism version 6.0 (GraphPad Software Inc., San Diego, CA, USA), statistical software packages. Statistical comparisons among groups were made using unpaired Student’s *t*-test. Multiple intergroup comparisons were made through one-way analysis of variance (ANOVA), followed by a *post hoc* Bonferroni multiple comparison test. Statistical significance was set at a *p* value of < 0.05 for all analyses.

## 3. Results

### 3.1. Doxycycline Exposure before C. difficile Infection

The mice with doxycycline exposure had higher cecum weight (1.3 ± 0.1 g vs. 0.5 ± 0.1 g; *p* < 0.001) and less body weight reduction (0.7 ± 0.5 g vs. −17.4 ± 0.2 g; *p* < 0.001) than the control mice, in which PBS was administered intraperitoneally (Figure 2A,B). Furthermore, doxycycline-exposed mice had higher expression levels of OCLN (1.4 ± 0.1 vs. 0.6 ± 0.2, *p* < 0.01) and IL-18 (1.85 ± 0.2 vs. 1.0 ± 0.2, *p* = 0.05) and lower expression levels of IL-1β (0.2 ± 0.04 vs. 1.0 ± 0.4, *p* = 0.04) in colonic tissues than control mice (Figure 2C).

Characters of epithelial inflammation, including neutrophil infiltration, epithelial damage, and irregular shape of colon mucosa, were noted in the pathology control (PBS) (red arrow in Figure 3). In contrast, in doxycycline-exposure mice, these epithelial inflammation characters were not obvious. After calculating the pathologic score, fewer neutrophil infiltrations (1.8 ± 0.8 vs. 4.4 ± 0.9/HPF, *p* = 0.001) and lower pathologic scores (2.4 ± 0.5 vs. 4.3 ± 1.3, *p* = 0.02) were observed in the colonic tissues of doxycycline-exposed mice than in control mice (Figure 3).

### 3.2. Therapeutic Effect of Oral Doxycycline on C. difficile Infection

While the animal was having diarrhea, due to the continuity discharge of gut contents, the cecum weight and body weight significantly decreased. All *C. difficile* infected mice, compared to mice without infection, showed symptoms of CDI, including loss of cecum weight at day 2 (Figure 4).

The *C. difficile*-infected mice treated with oral doxycycline, metronidazole, or vancomycin had fewer body weight changes than the untreated mice (1.1 ± 0.1 g, 1.3 ± 0.2 g, 1.2 ± 0.1 g, vs. 2.9 ± 0.3 g, all *p* < 0.001, respectively). Moreover, there were no differences in terms of body weight changes among the mice treated with doxycycline therapy and those treated with metronidazole (*p* = 0.44) or vancomycin therapy (*p* = 0.82) (Figure 4). Similarly, the cecum weights of infected mice treated with oral doxycycline or vancomycin therapy were higher than those of untreated mice (1.0 ± 0.1 g and 1.0 ± 0.1 g vs. 0.4 ± 0.03 g, *p* < 0.0001, respectively), but there were no differences between those of mice treated with metronidazole and untreated mice (0.6 ± 0.1 g vs. 0.4 ± 0.03 g; *p* = 0.15). The fecal *C. difficile* bacterial load was similar among mice treated with oral doxycycline, metronidazole, or vancomycin but lower than the control (PBS) mice.

Regarding inflammatory cytokines, significantly lower expression levels of MIP-2 in colonic tissues were noted in the mice treated with doxycycline, metronidazole, or vancomycin than in the untreated mice (0.4 ± 0.1, 0.5 ± 0.1, 0.2 ± 0.02 vs. 2.9 ± 1.3, *p* = 0.02, 0.03, 0.04, respectively) (Figure 5).

The expression levels of IL-18 were similar in the three treatment groups and untreated mice (*p* = 0.35). In terms of tight junction proteins, the expression level of ZO-1 in colonic tissue from the doxycycline-treated mice was higher than that in untreated mice (1.2 ± 0.1 vs. 0.8 ± 0.1, *p* = 0.02), and the expression levels of occludin in the doxycycline- or vancomycin-treated mice were higher than those in the untreated mice (1.4 ± 0.1 or 1.5 ± 0.2 vs. 0.5 ± 0.1, *p* < 0.0001).

Characters of epithelial inflammation, including neutrophil infiltration, epithelial damage, and irregular shape of colon mucosa, were noted in the pathology control (PBS) (red arrow in Figure 6). In contrast, for doxycycline-, vancomycin-, or metronidazole-treated mice, these epithelial inflammation characters were not obvious. After calculating the pathologic score, fewer neutrophil infiltrations (mean neutrophil number: 2.4 ± 0.5, 2.6 ± 0.9, 2.2 ± 0.8 vs. 5.2 ± 1.3, *p* < 0.001) per HPF and lower pathologic scores (2.2 ± 0.8, 2.4 ± 0.5, 2.6 ± 0.5 vs. 6.2 ± 1.5, *p* < 0.001) were noted in the doxycycline-, vancomycin-, or metronidazole-treated mice than in the untreated mice. However, there were no differences among the three antibiotic-treated groups (Figure 6).

## 4. Discussion

In the “Concurrent doxycycline exposure and *C. difficile* challenge in mice” part in our study, doxycycline exposure did not predispose mice to the development of CDI. This result is compatible with the clinical finding, suggesting that doxycycline protects against the development of CDI [17]. In a retrospective, case-control study of 1142 cases of hospital-acquired CDI from 1999 to 2005, doxycycline was associated with a reduced risk of CDI (odds ratio, 0.41) [15]. In another study, doxycycline receipt was associated with a 27% decrease in the incidence of CDI [16]. Of note, the intraperitoneal administration of doxycycline was concurrent with the oral challenge of vegetative *C. difficile* cells in our study. Therefore, the reduced disease severity of CDI in doxycycline-exposed mice may be due to its inhibitory ability against *C. difficile* and anti-inflammatory effects [26,27,28]. Almost all classes of antimicrobial agents have been associated with the disruption of normal colonic flora, predisposition to *C. difficile* colonization, and infection [29,30,31,32]. Most antibiotics decrease the density of intestinal bacteria and disrupt the intestinal microbiota. Such a disturbing effect on the microbiota may last for more than 8 weeks after the cessation of antibiotics [33]. Furthermore, exposure to more than one class of antibiotics had a higher rate of *C. difficile* colonization [34]. Although doxycycline has been reported to induce dysbiosis and decrease diversity in gut microbiota [27,28], concurrent exposure to doxycycline with *C. difficile* challenge in our study may lessen the influence of microbiota change induced by doxycycline on the development of CDI. Therefore, based on our study findings and clinical reports [15,16,17], doxycycline exposure did not aggravate the disease severity of CDI.

Furthermore, in the “Therapeutic effect of oral doxycycline on *C. difficile* infection” part in our study, concurrent intraperitoneal administration of doxycycline at the time of gastrointestinal challenge of *C. difficile* alleviates the disease severity of CDI through inhibiting fecal *C. difficile* bacterial load. The measurement of fecal *C. difficile* bacterial loads in our study support that the intraperitoneal doxycycline revealed antibacterial activity. Such a finding provides further in vivo evidence to suggest the application of doxycycline for the treatment of CDI in humans. The resistance rates of doxycycline in *C. difficile* isolates from animal products, animal stools, or soil were regarded to be low, ranging from 0% to 2.1% [35,36,37]. In a clinical setting, according to the multicenter study of 1112 clinical *C. difficile* isolates in Taiwan, the doxycycline resistance rate ranged from 7.7% to 17.3% in five hospitals [20]. The fecal *C. difficile*-inhibiting effect in a mice model and relatively low resistance rate among clinical *C. difficile* isolates make doxycycline a good surrogate for therapy for CDI.

The therapeutic role of doxycycline for CDI showed not only antimicrobial effects but also anti-inflammatory effect in our study. Doxycycline had been regarded as an antibiotic with anti-inflammatory, antioxidant, anti-apoptotic, and immunomodulatory effects [13]. Doxycycline may be beneficial for *C. difficile*-induced inflammation, which was evidenced by reports that *C. difficile* toxin could enhance the production of MIP-2 and TNF-α in macrophages to drive colonic inflammation, and the anti-inflammatory effect of doxycycline can enhance its therapeutic role in acute CDI [38,39,40,41]. In the polymicrobial infection sepsis mice model, doxycycline could ameliorate systemic and pulmonary inflammation through reducing the plasma and lung pro-inflammatory cytokines [40]. In a mice macrophages’ ex vivo study, doxycycline revealed an inhibitory effect on the production of IL-1β when stimulated with *Leptospira interrogans* or lipopolysaccharide [41]. The C57BL/6 mouse study by Hansen et al. showed that doxycycline administration decreased the transcription of inflammatory genes, including IL-1β, IL-10, IL-18, CXCL-1, MIP-2, and tissue necrosis factor (TNF) in the ileum and IL-18 in the colon [26]. These results were in accordance with the result of our study: Lower expression levels of inflammatory cytokines, such as MIP-2, and higher expression levels of tight junction proteins, such as ZO-1, were evident in mice treated with doxycycline. Thus, the net effect of oral doxycycline therapy for CDI was similar to that of metronidazole or vancomycin therapy, which was probably related to both the anti-inflammatory and antibacterial effects of doxycycline. The therapeutic role of oral doxycycline alone or in combination with metronidazole or vancomycin warrants more animal or clinical investigations.

There were some limitations in our study. First, the therapeutic effects of doxycycline on different *C. difficile* ribotypes were not studied. As mentioned, *C. difficile* RT078 lineage isolates had higher doxycycline MICs, and the therapeutic effect of doxycycline against CDI caused by the former isolates needs further evaluation. Second, a pressing issue of the current CDI therapy is the high recurrence rate after the end of antimicrobial therapy; CDI recurrence and the impact on gut microbiota by the doxycycline-containing regimen were not analyzed. Third, one of the difficulties in the treatment of CDI was recurrence; we did not assess the efficacy of doxycycline therapy for recurrent CDI. Fourth, doxycycline exposure was concurrent with oral *C. difficile* challenge in our study; the clinically unmet need for preventing the development of CDI through antibiotic prophylaxis was not evaluated here. Finally, oral doxycycline, metronidazole, or vancomycin was administered twice for only 1 day, but not for 10 days as the clinical recommendation for the treatment of CDI. A longer duration of doxycycline therapy may lead to a better therapeutic or pathogenic outcome and warrants further animal and clinical investigations.

In conclusion, our mouse model of CDI suggests that concurrent intraperitoneal administration of doxycycline and oral *C. difficile* challenge does not aggravate the disease severity of CDI. The anti-inflammatory effect and antibacterial activity of doxycycline may render it a potential therapeutic option in the treatment of CDI.

## Data Availability

No dataset is available.

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
