# Peer review of "Effect of Doxycycline in Decreasing the Severity of Clostridioides difficile Infection in Mice"

_antibiotics, 2022, doi:10.3390/antibiotics11010116_

Round 1

Reviewer 1 Report

The authors tested the effectiveness of Doxycycline in a Clostridioides difficile Infection (CDI) mouse model. They conclude that concurrent intraperitoneal administration of doxycycline and oral C. difficile challenge does not aggravate the disease severity of CDI, and oral doxycycline may be a potential therapeutic option for CDI.

The authors did a good job of citing the relevant literature and summarizing their results. However, their are a number of issues with their experimental design.

The greatest weakness of the manuscript was that the authors did not attempt to distinguish the mechanism of action by which Doxycycline reduced weight change and reduced inflammation in their CDI model. The most obvious answer would be that Doxycycline-treatment cleared the infection, or in the case of prophylactic use, prevented the establishment of an infection. However, the authors did not examine bacterial numbers at all in their study. These could have been examined relatively easily, given that the animals were killed for histopathology samples.

Relatedly, the authors mention in numerous places that Doxycycline possess both antibacterial and anti-inflammatory properties, yet they did not attempt to distinguish between these two properties in their model of CDI. Assuming Doxycycline's anti-inflammatory properties are responsible for the phenotypes the authors describe, there is no way of knowing whether these properties are C. dif -specific. A control demonstrating that Doxycycline reduces inflammation in a C. dif independent manner by using a completely different inflammatory agonist in the context of their model (LPS, CpG, etc) is a glaring omission.

Another omission in experimental design is that uninfected mice were not used as negative controls throughout the study; baseline body/cecum weights, cytokine production, neutrophil infiltration, etc. all lack a negative infection controls. It is also unclear what affect the four days of antibiotic cocktail treatment has on the abundance of native microbiota, as well as average mouse / cecum weight.

Author Response

Dear the editor of Antibiotics:

Enclosed, please find our revised manuscript entitled " Anti-Inflammation effect of doxycycline in decreasing the severity of Clostridioides difficile infection in mice ". We have considered carefully for all concerns that were raised by the reviewers. The manuscript was revised to a major extent in the Abstract, Materials and Methods, and Results, to improve the scientific logic and fluency of the whole manuscript. English editing has been done by an agent, American Journal Experts. The major changes were shown with “red words”. We hope that our changes can clarify the points raised.

We look forward to hearing from you.

Best Regards,

Pei-Jane Tsai, PhD

Center of Infectious Disease and Signaling Research, National Cheng Kung University, Tainan City, Taiwan

TEL: 886–6-2353535, ext. 5763 FAX: 886–6-2363956

E-mail: peijtsai@mail.ncku.edu.com

Yuan-Pin Hung, MD

Department of Internal Medicine, Tainan Hospital, Ministry of Health and Welfare, Tainan, Taiwan

TEL: 886–6-2200055, ext. 3125 FAX: 886–6-2200055, ext. 3125

E-mail: yuebin16@yahoo.com.tw

Wen-Chien Ko, MD

Division of Infectious Diseases, Department of Internal Medicine, National Cheng Kung University Hospital, No. 138, Sheng Li Road, Tainan, 70403, Taiwan

TEL: 886–6-2353535, ext. 3596 FAX: 886–6-2752038

E-mail: winston3415@gmail.com

Reviewers' comments:

Reviewer #1 (Comments for the Author):
The authors tested the effectiveness of Doxycycline in a Clostridioides difficile Infection (CDI) mouse model. They conclude that concurrent intraperitoneal administration of doxycycline and oral C. difficile challenge does not aggravate the disease severity of CDI, and oral doxycycline may be a potential therapeutic option for CDI.

The authors did a good job of citing the relevant literature and summarizing their results. However, their are a number of issues with their experimental design.

The greatest weakness of the manuscript was that the authors did not attempt to distinguish the mechanism of action by which Doxycycline reduced weight change and reduced inflammation in their CDI model. The most obvious answer would be that Doxycycline-treatment cleared the infection, or in the case of prophylactic use, prevented the establishment of an infection. However, the authors did not examine bacterial numbers at all in their study. These could have been examined relatively easily, given that the animals were killed for histopathology samples.
Reply:

Thanks for the suggestion. We agree the reviewer’s opinion that Doxycycline-treatment cleared the C. difficile infection. According to this, here we provided the evidence of the bacterial burden in feces. We found the bacterial loads of C. difficile in feces were decreased after doxycycline, metronidazole and vancomycin treatment compared to the mock control group. This result was correlated with CDI disease progression, which echoes the reviewer’s point of view. We added the operation procedure in Methods (line 134-137) and this results in Figure 4C and described on line 206-207.

Relatedly, the authors mention in numerous places that Doxycycline possess both antibacterial and anti-inflammatory properties, yet they did not attempt to distinguish between these two properties in their model of CDI. Assuming Doxycycline's anti-inflammatory properties are responsible for the phenotypes the authors describe, there is no way of knowing whether these properties are C. dif -specific. A control demonstrating that Doxycycline reduces inflammation in a C. dif independent manner by using a completely different inflammatory agonist in the context of their model (LPS, CpG, etc) is a glaring omission.

Reply:

We agree with the reviewer that we did not attempt to distinguish between anti-bacterial and anti-inflammation property of doxycycline. Although previous studies had demonstrated that doxycycline has both the antimicrobial property and the anti-inflammation feature (Singh, S.et al. Curr Mol Pharmacol 2021, doi:10.2174/1874467214666210210122628). In our study, we first demonstrated that doxycycline treatment in mice which administrated with 5-days antibiotic cocktail decreased the inflammatory cytokines, IL-1b and MIP-2, and increased the tight-junction protein, occludin. This anti-inflammatory effect is not C. difficile specific but supported that doxycycline treatment is not a risk predisposing factor for the development of CDI, which was known as clindamycin did. Following, our data supported that not only doxycycline but also metronidazole and vancomycin treatment, the inflammatory responses were all decreased after the clearance of C. difficile with the antibiotics that we provided. This antibacterial effect is C. difficile specific. Here, our results demonstrated that doxycycline treatment is not a risk factor for developing CDI but provides an antimicrobial property for an alternative therapeutic choice for CDI. For preventing the confusing our purpose, we revised our title as “Effect of doxycycline in decreasing the severity of Clostridioides difficile infection in mice”.

Another omission in experimental design is that uninfected mice were not used as negative controls throughout the study; baseline body/cecum weights, cytokine production, neutrophil infiltration, etc. all lack a negative infection controls. It is also unclear what affect the four days of antibiotic cocktail treatment has on the abundance of native microbiota, as well as average mouse / cecum weight.
Reply:

Thanks for the consideration. We provided the results of uninfected mice which administrated with 5-days antibiotic cocktail but treated with PBS as the negative control group in line 112-113 and Figure 2. In the treatment results, compared to the antibiotic treatment, the results of uninfected group were demonstrated by dash line (Figure 4). All C. difficile infected mice, compared to mice without infection showed symptoms of CDI, including loss of cecal weight at day 2 (line 193-194).

Reviewer 2 Report

The manuscript entitled  Anti-Inflammation Effect of Doxycycline in Decreasing the Se-verity of Clostridioides difficile Infection in Mice needs thorough re-writting before publication.

The topic is interesting and the results obtained have potential clinical application, but the work was written in a manner that is unclear and difficult for the reader. All parts require re-writing and completion. Likewise, figures whose contents are repeated (Fig 2A and 4A or 2B and 4B).

Moreover,

I don’t like a title.

Why the term “predisposing factor” has been added to keywords?

Introduction should be re-written. In present form it is chaotic without clearly indicated purpose of the study. Please add the information about inflammatory response and explain the need of anti-inflammatory treatment and the effect of some compounds. What is the reason of use vancomycin and fidaxomycin, please add the rationale to this paragraph. Generally, the Introduction looks as if part of the text has been cut off.

Materials and Methods

The study design is unclear. There is no information about the number of animals in particular groups, the number of animals used for the experiment and the approval of the ethical committee. Was the control group in this experiment? Please, clearly describe each of experimental group.

What was the marker of clinical severity of disorder?

Histopatological examination: When the animals were sacrificed?

Results:

Some procedures were omitted in Materials and Methods, please complete it.

The first sentence should be moved to Materials and Methods.

There is no information about clinical examination as well as body weight measurement, cecum weight, and evaluation of inflammatory response.

Why body weight and cecum weight were measured, please explain?

How the inflammatory response was assessed? This information lack in Introduction and Material and Methods parts.

I suggest that the key histopathological changes should be marked on the presented images or described in detail in the text.

Discussion is hard to read, please discuss obtained results with literature.

Author Response

Dear the editor of Antibiotics:

Enclosed, please find our revised manuscript entitled " Anti-Inflammation effect of doxycycline in decreasing the severity of Clostridioides difficile infection in mice ". We have considered carefully for all concerns that were raised by the reviewers. The manuscript was revised to a major extent in the Abstract, Materials and Methods, and Results, to improve the scientific logic and fluency of the whole manuscript. English editing has been done by an agent, American Journal Experts. The major changes were shown with “red words”. We hope that our changes can clarify the points raised.

We look forward to hearing from you.

Best Regards,

Pei-Jane Tsai, PhD

Center of Infectious Disease and Signaling Research, National Cheng Kung University, Tainan City, Taiwan

TEL: 886–6-2353535, ext. 5763 FAX: 886–6-2363956

E-mail: peijtsai@mail.ncku.edu.com

Yuan-Pin Hung, MD

Department of Internal Medicine, Tainan Hospital, Ministry of Health and Welfare, Tainan, Taiwan

TEL: 886–6-2200055, ext. 3125 FAX: 886–6-2200055, ext. 3125

E-mail: yuebin16@yahoo.com.tw

Wen-Chien Ko, MD

Division of Infectious Diseases, Department of Internal Medicine, National Cheng Kung University Hospital, No. 138, Sheng Li Road, Tainan, 70403, Taiwan

TEL: 886–6-2353535, ext. 3596 FAX: 886–6-2752038

E-mail: winston3415@gmail.com

Reviewers' comments:

Reviewer #2 (Comments for the Author):

The manuscript entitled  Anti-Inflammation Effect of Doxycycline in Decreasing the Se-verity of Clostridioides difficile Infection in Mice needs thorough re-writting before publication.

Reply:

Thanks for the consideration. Most of the parts were revised as suggested.

The topic is interesting and the results obtained have potential clinical application, but the work was written in a manner that is unclear and difficult for the reader. All parts require re-writing and completion. Likewise, figures whose contents are repeated (Fig 2A and 4A or 2B and 4B).

Reply:

Sorry for the confusing. Our study mainly included two part: in figure 2, we would like to emphasize whether doxycycline is a risk predisposing factor of developing CDI. In this study, we did not administration of C. difficile. We attempted to investigate the anti-inflammatory effect of doxycycline treatment in mice which administrated with 5-days antibiotic cocktail. Following, in figure 4, our data supported that not only doxycycline but also metronidazole and vancomycin treatment decreased the CDI disease symptoms after the clearance of C. difficile. We had revised the legend of figure 2 and figure 4 to avoid the misleading.

Moreover, I don’t like a title.

Reply:

To make it more persuasive, the title was revised as “Effect of doxycycline in decreasing the severity of Clostridioides difficile infection in mice”. Other parts were revised as suggested.

Why the term “predisposing factor” has been added to keywords?

Reply:

Sorry not to mention clear. This is because in our first main part, we attempted to emphasize whether doxycycline is a risk predisposing factor of developing CDI. Therefore, we used the “Doxycycline exposure before C. difficile infection” model to investigate whether doxycycline exposure as the risk predisposing factor to CDI. To avoid confusing, we had replaced “predisposing factor” with “exposure” to avoid the misleading.

Introduction should be re-written. In present form it is chaotic without clearly indicated purpose of the study. Please add the information about inflammatory response and explain the need of anti-inflammatory treatment and the effect of some compounds. What is the reason of use vancomycin and fidaxomycin, please add the rationale to this paragraph. Generally, the Introduction looks as if part of the text has been cut off.

Reply:

  1. The discussion of vancomycin and fidaxomicin was revised as: Vancomycin, besides direct anti-C. difficile effect had no anti-inflammatory effect. So in CDI mice model, vancomycin treatment was associated with reduced weight loss and diarrhea during acute infection (Li, Y. et al. BMC Infect Dis 2012, 12, 342). In contrast fidaxomicin and its primary metabolite OP-1118 were found to significantly inhibit difficile toxin A-mediated intestinal inflammation, and thus decreased tissue damage in the human colon (Koon, H.W. et al. Antimicrob Agents Chemother 2018, 62). So an antibiotics with both anti- C. difficile effect and anti-inflammation effect would be an excellent surrogate for treatment of CDI. (line 69-78)
  2. The association between inflammation and CDI treatment was revised as: Vancomycin, besides direct anti-C. difficile effect had no anti-inflammatory effect. So in CDI mice model, vancomycin treatment was associated with reduced weight loss and diarrhea during acute infection. However administration of vancomycin plus an anti-inflammatory agent, for example A2A adenosine receptor (A2AAR) agonist decreased inflammation and improved survival rates (Li, Y. et al. BMC Infect Dis 2012, 12, 342). (line 69-74)

Materials and Methods

The study design is unclear. There is no information about the number of animals in particular groups, the number of animals used for the experiment and the approval of the ethical committee. Was the control group in this experiment? Please, clearly describe each of experimental group.

Reply:

All mouse studies were with five mice in each group and repeated at least three times to confirm the results. This study had been approved by IACUC in National Cheng Kong University (Approval No. 102296) (line112-113 and 122-124). And the detail “INSTITUTIONAL REVIEW BOARD STATEMENT” was illustrated in line 306-311.

  1. The “vehicle group” using PBS (phosphate-buffered saline) was as the negative control group (line 109-110, and 122).
  2. For the “Concurrent doxycycline exposure and difficile challenge in mice” part, the two groups were: doxycycline (5 mg/kg), or PBS control (line 109-110)
  3. For the “oral antibiotic therapy for CDI in mice” part, the four groups were: one dose of doxycycline (125 mg/kg), metronidazole (250 mg/kg), vancomycin (50 mg/kg), or PBS control (line 121-122).

What was the marker of clinical severity of disorder?

Reply:

Thanks for the reminding. We added the descriptions in the line 127-132. Reported signs of colitis in mice included weight loss, and diarrhea. Based on previous studies, the changes of body weight, stool consistency, gross view of gut, and cecal weight, were selected to estimate the disease severity of CDI in mice (Johansson, M.E.et al. PLoS One 2010, 5, e12238 and Hung, Y.P. etal. J Infect Dis 2015, 212, 654-663).

Histopatological examination: When the animals were sacrificed?

Reply:

Thanks for reminding. All mice were sacrificed two days after infection and the colon tissues were examined for structural integrity by histological analysis (line 155).

Results: Some procedures were omitted in Materials and Methods, please complete it.

Reply:

Thanks for reminding. All the study design, clinical severity marker, and the timing of sacrifice were described clearly and completed as mentioned above in Materials and Methods.

The first sentence should be moved to Materials and Methods.

Reply:

Thanks for suggestion, we moved the first sentence to Materials and Methods (Line 113-114, and 123-124).

There is no information about clinical examination as well as body weight measurement, cecum weight, and evaluation of inflammatory response.

Reply:

Thanks for the reminding again. This part of description was revised in the line 127-132. Reported signs of colitis in mice included weight loss, and diarrhea. Based on previous studies, the changes of body weight, stool consistency, gross view of gut, and cecal weight, were selected to estimate the disease severity of CDI in mice (Johansson, M.E.et al. PLoS One 2010, 5, e12238 and Hung, Y.P. etal. J Infect Dis 2015, 212, 654-663).

Why body weight and cecum weight were measured, please explain?

Reply:

Sorry not mentioned clearly. While animal was under diarrhea, due to the continuity discharge of gut contents, the cecum weight and body weight significantly decreased. ( line 194-195)

How the inflammatory response was assessed? This information lack in Introduction and Material and Methods parts.

Reply:

Thanks for the reminding. We revised this part and described clearly in the Introduction and Material and Methods parts as followings:

The inflammatory responses in colon tissues were performed by histological analysis and inflammatory cytokines levels. The infiltration of immune cells was observed in histopathological analysis and the expression of inflammatory cytokines were analyzed by real-time PCR. All the methodology and description were:

  1. In Methodology: The markers of inflammation, including neutrophil infiltration, epithelial damage, and irregular shape of colon mucosa were analyzed.(line 168-169)
  2. In Result: Characters of epithelial inflammation, including neutrophil infiltration, epithelial damage, and irregular shape of colon mucosa were noted in the pathology control (PBS) (red arrow in Fig 3).(line 185-187) and Characters of epithelial inflammation, including neutrophil infiltration, epithelial damage, and irregular shape of colon mucosa were noted in the pathology control (PBS) (red arrow in Fig 6). (line 217-219)

I suggest that the key histopathological changes should be marked on the presented images or described in detail in the text.

Reply:

Thanks for the suggestion, we evaluated the histopathological images and marked the important phenoniums as following descriptions:

  1. Characters of Epithelial inflammation (neutrophil infiltration, epithelial damage, and irregular shape of colon mucosa) had been marked with red arrow on Figure 3 and 6.
  2. In “Doxycycline exposure before C. difficile infection” part, Characters of epithelial inflammation, including neutrophil infiltration, epithelial damage, and irregular shape of colon mucosa were noted in the pathology control (PBS) (red arrow in Fig 3). In contrast in doxycycline exposure mice, these epithelial inflammation characters was not obvious. (line 185-191)
  3. In “Therapeutic effect of oral doxycycline on C. difficile infection” part, Characters of epithelial inflammation, including neutrophil infiltration, epithelial damage, and irregular shape of colon mucosa were noted in the pathology control (PBS) (red arrow in Fig 6). In contrast doxycycline-, vancomycin-, or metronidazole-treated mice, these epithelial inflammation characters was not obvious. (line 217-224)

Discussion is hard to read, please discuss obtained results with literature.

Reply:

Thanks for the suggestion. The Discussion had been revised according to the results, including:

  1. Discussion related to “doxycycline exposure did not predispose mice to the development of CDI.” Was: In the “Concurrent doxycycline exposure and difficile challenge in mice” part in our study, doxycycline exposure did not predispose mice to the development of CDI. This result is compatible with the clinical finding suggesting that doxycycline protects against the development of CDI [17]. In a retrospective case-control study of 1,142 cases of hospital-acquired CDI from 1999 to 2005, doxycycline was associated with a reduced risk of CDI (odds ratio, 0.41) [15]. In another study, doxycycline receipt was associated with a 27% decrease in the incidence of CDI [16]. Of note, the intraperitoneal administration of doxycycline was concurrent with the oral challenge of vegetative C. difficile cells in our study. Therefore, the reduced disease severity of CDI in doxycycline-exposed mice may be due to its inhibitory ability against C. difficile and anti-inflammatory effects [26-28]. Almost all classes of antimicrobial agents have been associated with the disruption of normal colonic flora, predisposition to C. difficile colonization and infection [29-32]. Most antibiotics decrease the density of intestinal bacteria and disrupt the intestinal microbiota. Such a disturbing effect on the microbiota may last for more than eight weeks after the cessation of antibiotics [33]. Furthermore, exposure to more than one class of antibiotics had a higher rate of C. difficile colonization [34]. Although doxycycline has been reported to induce dysbiosis and decrease diversity in gut microbiota [27,28], concurrent exposure to doxycycline with C. difficile challenge in our study may lessen the influence of microbiota change induced by doxycycline on the development of CDI. Therefore, based on our study findings and clinical reports [15-17], doxycycline exposure did not aggravate the disease severity of CDI. (line 227-247)
  2. Discussion related to “challenge of difficile alleviates the disease severity of CDI through inhibiting fecal C. difficile bacterial load” was: Furthermore, in the “Therapeutic effect of oral doxycycline on C. difficile infection” part in our study, concurrent intraperitoneal administration of doxycycline at the time of gastrointestinal challenge of C. difficile alleviates the disease severity of CDI through inhibiting fecal C. difficile bacterial load. The measurement of fecal C. difficile bacterial loads in our study support that the intraperitoneal doxycycline revealed antibacterial activity. Such a finding provides further in vivo evidence to suggest the application of doxycycline for the treatment of CDI in humans. The resistance rates of doxycycline in C. difficile isolates from animal products, animal stools, or soil were regarded to be low, ranging from 0% to 2.1% [35-37]. In clinical setting according to the multicenter study of 1,112 clinical C. difficile isolates in Taiwan, the doxycycline resistance rate ranged from 7.7% to 17.3% in five hospitals [20]. The fecal C. difficile- inhibiting effect in mice model and relative low resistant rate among clinical C. difficile isolates makes doxycycline a good surrogate for therapy for CDI. (line 248-260)
  3. Discussion related to “Therapeutic role of doxycycline for CDI is not only antimicrobial effects but also anti-inflammatory effect in our study” was: Therapeutic role of doxycycline for CDI is not only antimicrobial effects but also anti-inflammatory effect in our study. Doxycycline had been regarded as an antibiotics with anti-inflammatory, antioxidant, anti-apoptotic, and immunomodulatory effects [13]. Doxycycline may be beneficial for difficile-induced inflammation, which was evidenced by reports that C. difficile toxin could enhance the production of MIP-2 and TNF-α in macrophages to drive colonic inflammation, and the anti-inflammatory effect of doxycycline can enhance its therapeutic role in acute CDI [38-41]. In the polymicrobial-infection sepsis mice model doxycycline could ameliorates systemic and pulmonary inflammation through reducing the plasma and lung pro-inflammatory cytokines [40]. In mice macrophages ex vivo study, doxycycline revealed inhibitory effect on the production of IL-1β when stimulated with Leptospira interrogans or lipopolysaccharide [41]. In the C57BL/6 mouse study by Hansen et al., which showed that doxycycline administration decreased the transcription of inflammatory genes, including IL-1β, IL-10, IL-18, CXCL-1, MIP-2, and tissue necrosis factor (TNF) in the ileum, and IL-18 in the colon [26]. These results were in accordance with the result of our study, lower expression levels of inflammatory cytokines, such as MIP-2, and higher expression levels of tight junction proteins, such as ZO-1, were evident in mice treated with doxycycline. Thus, the net effect of oral doxycycline therapy for CDI was similar to that of metronidazole or vancomycin therapy, which is probably related to both the anti-inflammatory and antibacterial effects of doxycycline. The therapeutic role of oral doxycycline alone or in combination with metronidazole or vancomycin warrants more animal or clinical investigations. (line 261-281)

Round 2

Reviewer 1 Report

The authors have adequately addressed my concerns.

Reviewer 2 Report

I appreciate changes done by Authors. However, still:

The sentence  “Vancomycin, besides direct anti-C. difficile effect had no anti-inflammatory effect”  is confusing. On the basis of the literature cited I suppose that Authors meant that “In a murine model of CDI, treatment with vancomycin caused decrease of weight loss and diarrhea during acute infection, but was involved in high recurrence and late-onset death, with higher overall mortality than in untreated infected animals, hence the need for additional anti-inflammatory treatment”. Am I right? Please write this part in a more understandable way.

“Why body weight and cecum weight were measured, please explain?”

By asking this question, I wanted to receive a justification for  these measurements for research purposes and the Authors have already replied in previous part.

I have not other comments

Author Response

Dear the editor of Antibiotics:

Enclosed, please find our revised manuscript entitled " Anti-Inflammation effect of doxycycline in decreasing the severity of Clostridioides difficile infection in mice ". We have considered carefully for all concerns that were raised by the reviewers. The manuscript was revised to a major extent in the Abstract, Materials and Methods, and Results, to improve the scientific logic and fluency of the whole manuscript. English editing has been done by an agent, American Journal Experts. The major changes were shown with “red words”. We hope that our changes can clarify the points raised.

We look forward to hearing from you.

Best Regards,

Pei-Jane Tsai, PhD

Center of Infectious Disease and Signaling Research, National Cheng Kung University, Tainan City, Taiwan

TEL: 886–6-2353535, ext. 5763 FAX: 886–6-2363956

E-mail: peijtsai@mail.ncku.edu.com

Yuan-Pin Hung, MD

Department of Internal Medicine, Tainan Hospital, Ministry of Health and Welfare, Tainan, Taiwan

TEL: 886–6-2200055, ext. 3125 FAX: 886–6-2200055, ext. 3125

E-mail: yuebin16@yahoo.com.tw

Wen-Chien Ko, MD

Division of Infectious Diseases, Department of Internal Medicine, National Cheng Kung University Hospital, No. 138, Sheng Li Road, Tainan, 70403, Taiwan

TEL: 886–6-2353535, ext. 3596 FAX: 886–6-2752038

E-mail: winston3415@gmail.com

Reviewers' comments:

I appreciate changes done by Authors. However, still:

The sentence “Vancomycin, besides direct anti-C. difficile effect had no anti-inflammatory effect”  is confusing. On the basis of the literature cited I suppose that Authors meant that “In a murine model of CDI, treatment with vancomycin caused decrease of weight loss and diarrhea during acute infection, but was involved in high recurrence and late-onset death, with higher overall mortality than in untreated infected animals, hence the need for additional anti-inflammatory treatment”. Am I right? Please write this part in a more understandable way. 

Reply: We agree with the suggestions from the reviewer. The sentence was revised as “In the mouse model of CDI, vancomycin therapy results in the mitigation of weight loss and diarrhea during acute infection, but is associated with CDI recurrence and late-onset death, indicative of the need for other therapeutic agents” (Line 72-74).

“Why body weight and cecum weight were measured, please explain?”

By asking this question, I wanted to receive a justification for these measurements for research purposes and the Authors have already replied in previous part.

Reply: In our published article about the mouse model of CDI (J Infect Dis 2015; 212: 654-663.), severe CDI was associated with frequent diarrhea, body weight loss, and less fecal content in cecum. The latter was reflected by a decrease in cecum weight. So in our study body weight and cecum weight are measured to reflect the disease severity of CDI in the mouse model. (Line 109-112).
